# R-Spondin chromosome rearrangements drive Wnt-dependent tumour initiation and maintenance in the intestine

Teng Han[1,2], Emma M. Schatoff[1,3], Charles Murphy[1,2,4], Maria Paz Zafra[1], John E. Wilkinson[5], Olivier Elemento[1] & Lukas E. Dow[1,2,6]

Defining the genetic drivers of cancer progression is a key in understanding disease biology and developing effective targeted therapies. Chromosome rearrangements are a common feature of human malignancies, but whether they represent bona fide cancer drivers and therapeutically actionable targets, requires functional testing. Here, we describe the generation of transgenic, inducible CRISPR-based mouse systems to engineer and study recurrent colon cancer-associated *EIF3E–RSPO2* and *PTPRK–RSPO3* chromosome rearrangements *in vivo*. We show that both *Rspo2* and *Rspo3* fusion events are sufficient to initiate hyperplasia and tumour development *in vivo*, without additional cooperating genetic events. *Rspo*-fusion tumours are entirely Wnt-dependent, as treatment with an inhibitor of Wnt secretion, LGK974, drives rapid tumour clearance from the intestinal mucosa without effects on normal intestinal crypts. Altogether, our study provides direct evidence that endogenous *Rspo2 and Rspo3* chromosome rearrangements can initiate and maintain tumour development, and indicate a viable therapeutic window for LGK974 treatment of RSPO-fusion cancers.

[1] Department of Medicine, Sandra and Edward Meyer Cancer Center, Weill Cornell Medicine, New York, New York 10021, USA. [2] Weill Cornell Graduate School of Medical Sciences, Weill Cornell Medicine, New York, New York 10021, USA. [3] Weill Cornell/Rockefeller/Sloan Kettering Tri-I MD-PhD Program, New York, New York 10021, USA. [4] The Tri-Institutional Training Program in Computational Biology and Medicine, New York, New York 10021, USA. [5] Department of Pathology, University of Michigan School of Medicine, Ann Arbor, Michigan 48109, USA. [6] Department of Biochemistry, Weill Cornell Medicine, New York, New York 10021, USA. Correspondence and requests for materials should be addressed to L.E.D. (email: lud2005@med.cornell.edu).

Wnt pathway hyperactivation is a near ubiquitous feature of colorectal cancer (CRC), most frequently caused by truncating mutations in the adenomatous polyposis coli (APC) tumour suppressor. Numerous studies have demonstrated that APC loss and/or WNT pathway activation can initiate tumour development in the mammalian intestine[1–5], and we recently showed that re-engaging endogenous Wnt regulation via Apc restoration is sufficient to induce complete and sustained tumour regression, even in cancers carrying potent oncogenic mutations in Kras and p53 (ref. 6). Thus, hyperactive WNT signalling is the predominant oncogenic driver in CRC. However, it is not clear which, if any, genetic alterations drive tumour initiation in the 10–15% of CRCs that do not carry APC or downstream WNT pathway mutations.

Chromosomal rearrangements are a common feature of human malignancies and include translocations, duplications, inversions and deletions. These large-scale genetic events can result in the production of driving oncogenes and therapeutically actionable targets in multiple tumour types. Recently, de Sauvage and colleagues[7] described the first examples of recurrent chromosome rearrangements in CRC, involving members of the R-Spondin (RSPO) family, RSPO2 and RSPO3. In both cases, the rearrangements result in fusion transcripts, EIF3E–RSPO2 and PTPRK–RSPO3 that drive marked overexpression of the downstream RSPO gene.

RSPOs are secreted proteins that act as ligands for the leucine-rich repeat containing G-protein coupled receptor (LGR) family (LGR4/5/6). RSPO binding to LGRs potentiates WNT signalling by sequestering the E3 ubiquitin ligases RNF43 and ZNRF3, and prevents the turnover of frizzled (FZD)/LRP-receptor complexes at the cell membrane[8]. All reported RSPO-fusion CRCs are mutually exclusive of APC mutations[7,9], suggesting RSPO rearrangements may be a key genetic driver in CRC. However, while RSPOs are potent WNT signalling agonists, they only induce marked WNT pathway activation in the presence of WNT ligand[7,10]. Thus, it is not clear whether endogenous RSPO rearrangements alone are sufficient to induce tumorigenesis in the intestine.

Most gain- and loss-of-function mutations observed in human cancers can be easily modelled in mice for functional studies, while chromosomal rearrangements have proven challenging to faithfully recapitulate using traditional in vivo modelling tools. CRISPR/Cas9 genome editing provides an opportunity to quickly induce gross genomic events. Thus, we adapted an inducible CRISPR (iCRISPR) transgenic platform[11] to engineer Rspo2 and Rspo3 rearrangements in vivo in the mouse intestine. Using this system, we provide direct evidence that endogenous EIF3E–RSPO2 and PTPRK–RSPO3 fusion events are sufficient to initiate tumour development in the intestine. Importantly, while P-Rspo3 fusion tumours are morphologically similar to Apc-mutant adenomas, transcriptionally they more closely resemble normal stem and progenitor cells, and are exquisitely sensitive to inhibitors of Wnt ligand secretion.

## Results

**RSPO fusions support the growth of intestinal epithelium.** To understand the contribution of RSPO fusions to tumorigenesis in the gut, we set out to determine the impact of these alterations in genetically defined model systems. In the normal intestine, RSPOs are produced by resident stromal cells to support epithelial stem cell growth and regeneration[12]. In fact, RSPO is a key factor required for the growth of almost all epithelial organoid cultures[13–16]. To first determine whether Rspo2 or Rspo3 production from epithelial cells could provide a cell-intrinsic growth advantage, we transduced wildtype mouse intestinal organoids with retroviruses expressing murine Rspo2, Rspo3, and the EIF3E–RSPO2 and PTPRK–RSPO3 fusion transcripts linked to GFP (Supplementary Fig. 1a). Neomycin-selected organoid cultures showed high levels of transgene expression, and looked indistinguishable from controls (Supplementary Fig. 1b,c). While non-transduced cells could not proliferate in the absence of exogenous RSPO1, expression of Rspo2, Rspo3 or either of the fusion transcripts, enabled indefinite propagation in the absence of this growth factor, confirming that Rspo2 or Rspo3 produced from intestinal epithelium is sufficient to support ongoing stem-cell maintenance and proliferation.

**Generation of RSPO fusions using CRISPR/Cas9.** Heterologous cDNA overexpression systems often yield spurious phenotypes by driving supraphysiological levels of protein expression that do not mimic the native normal or tumour conditions, as has been well documented for oncogenic KRAS models[17,18]. To directly examine the impact of each RSPO-fusion on intestinal epithelium, we set out to precisely recreate the chromosome rearrangements in mice using CRISPR/Cas9 genome editing. The reported EIF3E–RSPO2 (hereafter, E-RSPO2) rearrangement is an intrachromosomal deletion between intron 1 of EIF3E and intron 1 of RSPO2, while the PTPRK-RSPO3 (P-RSPO3) rearrangement is an inversion between intron 1 (or intron 6) of PTPRK and intron 1 of RSPO3 (Fig. 1a). To identify pairs of single guide RNAs (sgRNAs) that could induce simultaneous DNA breaks and generate the appropriate rearrangements in the mouse genome, we screened several combinations of sgRNAs targeting the first intron of each gene, and chose two pairs that produced an expected fusion amplicon (Supplementary Fig. 2). We delivered each tandem sgRNA combination, in a doxycycline (dox)-inducible Cas9 lentiviral construct (L3CGP), into rtTA-expressing 3T3 cells, and treated with dox to induce Cas9 expression. After only 2 days of dox treatment, we could detect the expected chromosome rearrangements at both the genomic loci (Fig. 1b), and as expressed RNA fusion transcripts (Fig. 1c).

Two previous efforts to model chromosome rearrangements in vivo in the lung have exploited the accessibility of this tissue to deliver CRISPR components in viral vectors[19,20]. Murine intestinal epithelium is not easily transduced in vivo, so instead we turned to an inducible transgenic platform we recently described[11], to produce mice carrying both specific sgRNAs, and a dox-regulated Cas9 transgene. For this, we shuttled each validated sgRNA combination into the c3GIC9 Col1a1-targeting vector (Fig. 2a), and generated transgenic KH2 embryonic stem cells (ESCs)[21,22]. Treatment of correctly targeted ESC clones with dox for 4 days induced expected chromosome rearrangements and expression of the Rspo-fusion transcripts (Fig. 2b, Supplementary Fig. 3a). Importantly, we could detect fusion transcripts for both E-Rspo2 and P-Rspo3 using two independent sets of sgRNAs, but not with 'no sgRNA' control clones (Fig. 2b), confirming the generation of these genetic events was not due to non-specific activation of the Cas9 endonuclease. Finally, we screened the top five predicted off-target sites for each sgRNA, and were unable to detect any non-specific genomic cleavage (Supplementary Table 1; Supplementary Fig. 4).

To further ensure the fidelity of each genomic rearrangement, we performed multi-colour DNA fluorescence in situ hybridization (DNA FISH) on dox-treated ESC clones using internal and flanking probes. Two-colour FISH staining for Eif3e and Rspo2 showed expected loss of the 3′ Eif3e probe, while 3-colour FISH confirmed the P-Rspo3 rearrangement (Fig. 2c). Finally, as a functional readout of the chromosome alterations, quantitative RT-PCR (qRT-PCR) for Rspo2 and Rspo3 showed marked upregulation of each gene only in clones harbouring the associated transcript fusion (Supplementary Fig. 3b).

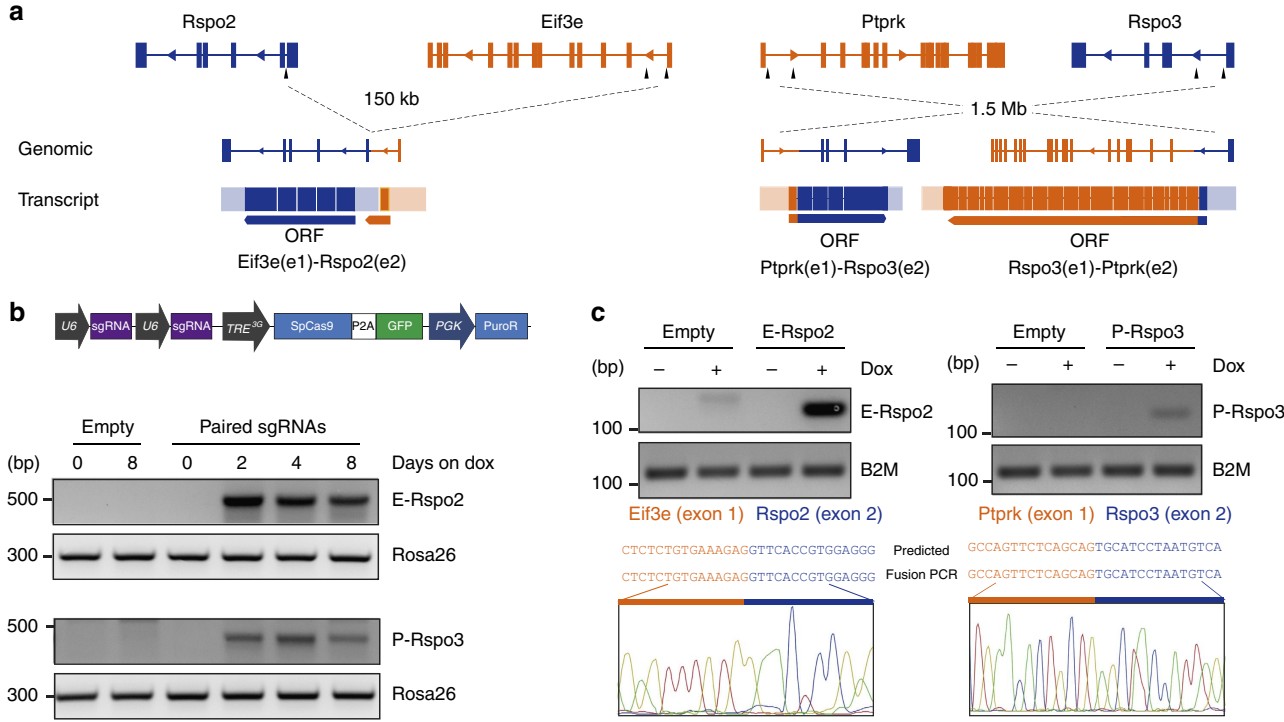

**Figure 1 | Induction of *EIF3E–RSPO2* and *PTPRK–RSPO3* fusions using inducible CRISPR.** (**a**) Schematic representation of chromosomal rearrangements involving *Rspo2* and *Eif3e* (left), and *Ptprk* and *Rspo3* (right). (**b**) Dox-inducible lentiviral vector (upper). Paired U6-sgRNAs are cloned into the vector upstream of TRE³ᴳ promoter. Detection of *EIF3E–RSPO2* (*E-Rspo2*) and *PTPRK–RSPO3* (*P-Rspo3*) genomic rearrangements by fusion-specific PCR, on genomic DNA extracted from puromycin-selected 3T3 cells at multiple time points (lower). (**c**) Detection of the *E-Rspo2* and *P-Rspo3* fusion transcripts using fusion-specific PCR primers on cDNA from day 4 dox-treated 3T3 cells in **b**.

**E-Rspo2 fusions do not enable RSPO1-independent growth.** We next generated transgenic mice by blastocyst injection, and derived *R26-rtTA/c3GIC9* bi-transgenic intestinal organoids. Analysis of gDNA from the bulk, dox-treated population, confirmed the presence of the *E-Rspo2* and *P-Rspo3* rearrangements, as well as upregulation of Rspo2 and Rspo3, respectively (Fig. 3a; Supplementary Fig. 5a,b). We cultured dox-treated and dox-naive organoids in basal media containing only EGF and Noggin (EN), and surprisingly, in contrast to what we observed with the *Rspo2* cDNA, dox-treated *E-Rspo2* organoids could not survive in the absence of exogenous RSPO1 (Supplementary Fig. 5c). We repeated these experiments on multiple independent mice (*n* = 4), and despite confirming the presence of the rearrangement following dox treatment, we were unable to generate RSPO1-independent *E-Rspo2* organoids. As the *E-Rspo2* rearrangement produces a fusion transcript, but not a fusion protein, it is unlikely that changes in protein folding or membrane targeting could explain the failure to induce RSPO1 independence. However, in both mice and humans, the *E-Rspo2* chimeric mRNA transcript generates an upstream, truncated open reading frame (ORF) from the *Eif3e* exon 1 that could influence translation of the downstream *Rspo2* ORF (Supplementary Fig. 5d). We were unable to reliably detect endogenous Rspo2 protein, and so developed a surrogate fluorescent reporter to measure how the *EIF3E–RSPO2* fusion impacts the level of (Rspo2) protein expression. For this, we generated two retroviral vectors expressing tdTomato immediately downstream of either the endogenous *Rspo2* (exon 2) untranslated region (UTR), or the chimeric *EIF3E–RSPO2* UTR (Supplementary Fig. 5e). We transduced 3T3 cells and quantified mean fluorescence intensity relative to the expression of each transcript, measured by qPCR. In this assay, the presence of the *EIF3E–RSPO2* chimeric UTR reduced the translation of tdTomato 10-fold (*n* = 3, *P* < 0.0001; Supplementary Fig. 5e).

Thus, while the E-Rspo2 fusion drives high levels of Rspo2 transcript, protein production is limited by an upstream, truncated, *Eif3e* ORF. Interestingly, in all cases identified, *EIF3E–RSPO2* rearrangements in human CRC are coincident with amplification of fusion locus on chromosome 8q[7]. This perhaps suggests that both rearrangement and increased dosage of the fusion are required for RSPO2-mediated cell transformation. This notion is consistent with our data showing that enforced Rspo2 cDNA overexpression drives growth factor independence, but the endogenous fusion does not.

**P-Rspo3 fusions do not induce major changes in organoids.** In contrast to *E-Rspo2* organoids, dox-treated *P-Rspo3* organoids showed robust polyclonal expansion following RSPO1 withdrawal, and maintained expression of stem-cell markers when cultured in EN media, in contrast to wildtype organoids (Supplementary Fig. 6). Although RSPO1-independent, *P-Rspo3* organoids were morphologically indistinguishable from wildtype cultures grown in complete ENR media (Fig. 3b; Supplementary Movies 1–4). The similarity of *P-Rspo3* organoids (in EN) and wildtype organoids (in ENR), prompted us to ask whether RSPO1-independent cells did in fact contain the *P-Rspo3* rearrangement, or were maintained by local paracrine signalling from *P-Rspo3*-positive cells secreting the ligand. Analysis of multiple independent *P-Rspo3* organoid cultures using a fusion-specific genomic Taqman assay, indicated that the majority, if not all of the cells in the culture carried a *P-Rspo3* rearrangement (Supplementary Fig. 7a). However, as this assay measures the frequency of the *P-Rspo3* rearrangement in the bulk population, it is possible that the presence of homozygous inversions in some cells could mask the presence of wildtype crypts. To more directly assess whether *P-Rspo3* fusion cells are capable of supporting the growth of adjacent wildtype cells, we

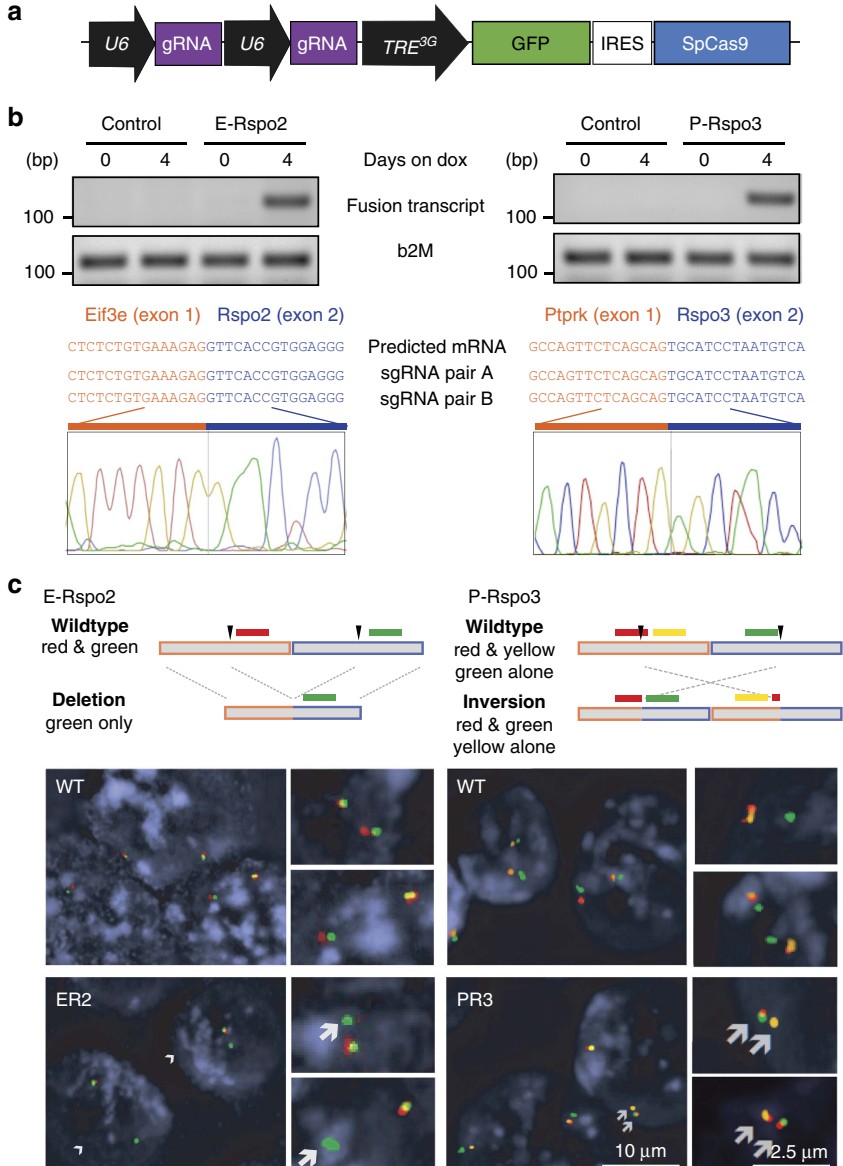

**Figure 2 | Induction of *E-Rspo2* and *P-Rspo3* fusions in transgenic iCRISPR ESCs. (a)** Schematic of iCRISPR *col1a1*-targeting vector. **(b)** Detection of the *E-Rspo2* and *P-Rspo3* fusion transcripts using fusion-specific PCR primers on cDNA from targeted ESC clones, 4 days post doxycycline treatment (upper). Sanger sequencing of the PCR product shows the expected splice junctions in expressed RNA transcripts (lower). **(c)** Details of the DNA FISH strategies to detect the *E-Rspo2* deletion, and *P-Rspo3* inversion (upper). Multi-colour DNA FISH on WT and isogenic *E-Rspo2* or *P-Rspo3* ESC clones. Rearranged alleles are highlighted with white arrows (lower).

co-cultured independent *P-Rspo3* organoid lines ($n = 2$), with GFP-positive, wildtype crypts and monitored GFP fluorescence over time (Supplementary Fig. 7b). In the presence of exogenous RSPO1 (ENR), neither cell population showed a growth advantage over 10 days in culture. In contrast, in RSPO1-free media (EN), P-Rspo3 organoids were able to proliferate and expand, while GFP-positive wildtype organoids were rapidly depleted (Supplementary Fig. 7b). Altogether, these data suggest that the *P-Rspo3* rearrangement provides a cell-intrinsic benefit for organoid growth, but is not sufficient to act as a paracrine factor to support the growth of adjacent, unmodified cells.

Given the robust growth of *P-Rspo3* organoids, and the uniformity of the chromosomal rearrangement throughout the culture, we next asked how the *P-Rspo3* inversion alters the behaviour of these otherwise genetically normal intestinal epithelial cells. *P-Rspo3* organoids developed crypt budding structures at a similar rate (Fig. 3b; Supplementary

Movies 1–4), and in contrast to $Apc^{\Delta/\Delta}$ spheroids, showed a regular pattern of EdU-positive (stem and progenitor) and lysozyme-positive (Paneth) cells in the crypt projections, and differentiated cells (Krt20) in the central villus domain (Fig. 3c). Likewise, the central lumen of the organoids displayed strong alkaline phosphatase activity, indicating the presence of functional enterocytes (Fig. 3c). As expected, *P-Rspo3* organoids showed a significant upregulation of *Rspo3* (3,500-fold), and reciprocal downregulation of the fusion partner, *Ptprk* (Fig. 3d). We did not detect substantial changes in expressed transcripts that lie within the inversion segment (Supplementary Fig. 8). Interestingly, while Rspo is a potent Wnt agonist, we did not observe any changes in classic Wnt-responsive genes (Fig. 3d), in contrast to a recent study using a CAGs-driven Rspo3 transgene[23].

To assess the impact of the *P-Rspo3* rearrangement more globally, we performed transcriptome profiling by RNAseq,

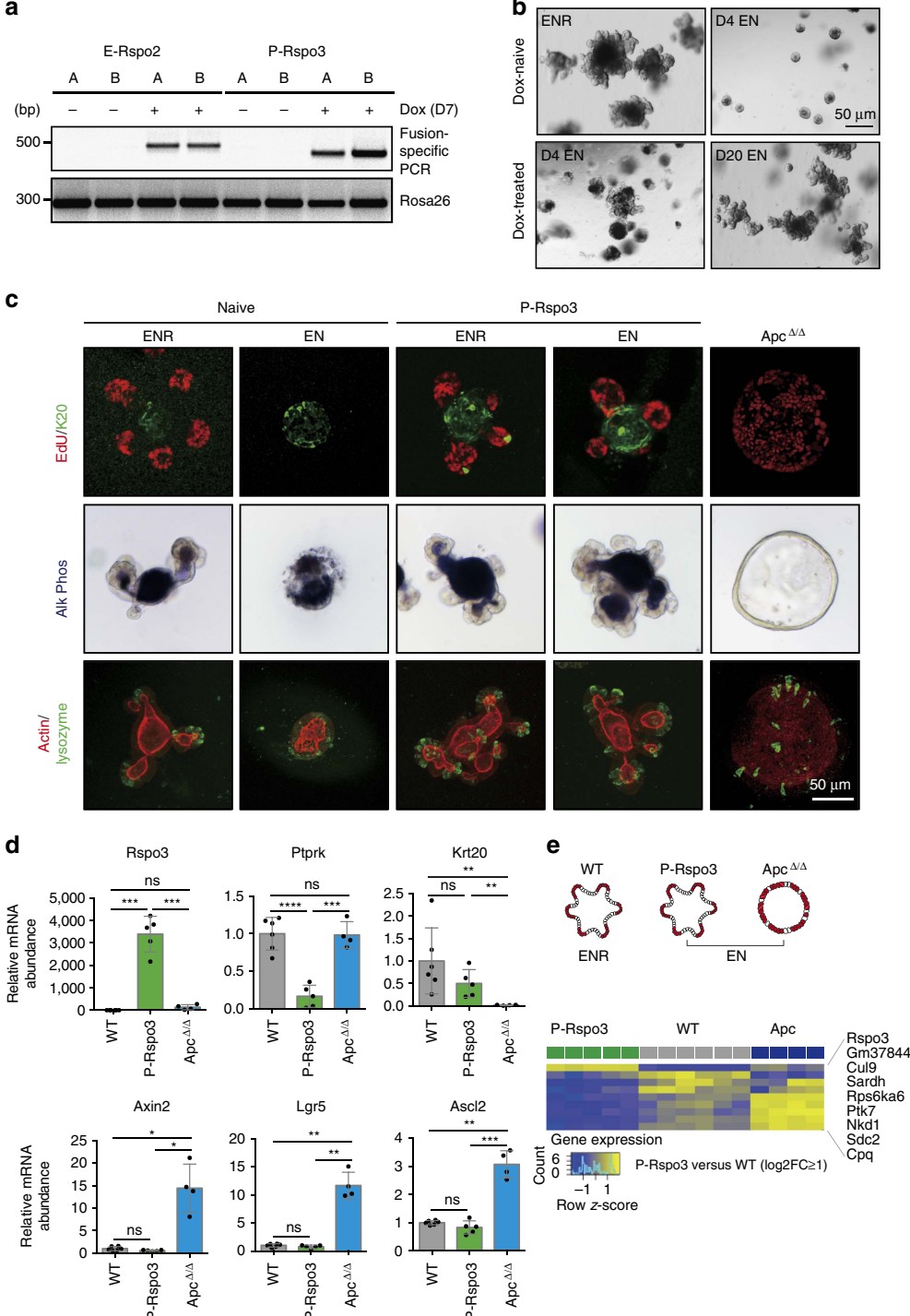

**Figure 3 | Intestinal organoids carrying the Ptprk–Rspo3 fusion are RSPO1-independent.** (**a**) Detection of E-Rspo2 and P-Rspo3 rearrangements using fusion-specific PCR primers on genomic DNA extracted from c3GIC9-E-Rspo2 and c3GIC9-P-Rspo3 mouse small intestinal organoids, 7 days post doxycycline treatment. (**b**) Bright-field images of dox-naive and dox-treated P-Rspo3 organoids cultured in ENR or EN medium, as indicated. Scale bar, 50 µm. (**c**) Confocal immunofluorescent images of dox-naive, P-Rspo3 and Apc-deleted organoids cultured in ENR and EN medium, as indicated, showing markers of proliferation (EdU), differentiation (K20 and alkaline phosphatase activity) and Paneth cells (lysozyme). Scale bar, 50 µm. (**d**) Graphs represent qRT-PCR results of Rspo3, Ptprk, K20 and Wnt target genes (Axin2, Lgr5 and Ascl2) on WT, P-Rspo3 and Apc-deleted organoids ($n \geq 4$, bars represent mean values $+/-$ s.d., *$P < 0.05$, **$P < 0.01$, ***$P < 0.001$, two-sided $t$-test with Welch correction). (**e**) Schematic of culture conditions of WT, P-Rspo3 and Apc-deleted organoids for RNAseq (upper). Heatmap indicates up (yellow) and downregulated (blue) transcripts (log2FC$\geq$1) in P-Rspo3 organoids (grey) compared to WT/naive cultures (green) (lower).

compared to wildtype/naive organoids and $Apc^{\Delta/\Delta}$ spheroids. To avoid the identification of gene signatures that simply report the differentiation of wildtype organoids in the absence of RSPO1, we collected RNA from wildtype/dox-naive cells in complete ENR, and from *P-Rspo3* and $Apc^{\Delta/\Delta}$ cultures in RSPO-free conditions (EN) (Fig. 3e). $Apc^{\Delta/\Delta}$ organoids showed a markedly altered

transcriptional profile, with 5,294 genes showing significant differential expression changes (log2FC > 1; 2,598 upregulated and 2,696 downregulated, adj. P-value < 0.05) (Supplementary Fig. 9a; Supplementary Data 1,2). As expected, differentially expressed transcripts were enriched for the Wnt pathway and colorectal cancer-associated genes (Supplementary Fig. 9b). Remarkably, P-Rspo3 organoids showed only a single gene upregulated (Rspo3), and eight significantly downregulated genes (Fig. 3e; Supplementary Fig. 10, Supplementary Data 3,4). While two of the eight downregulated genes are associated with the Wnt pathway regulation (Nkd1 and Ptk7)[24,25], perhaps suggesting low-level transcriptional feedback on the pathway, any such regulation seemingly has little impact on the overall transcriptional output.

In all, these organoid data show that while the PTPRK–RSPO3 rearrangement enables RSPO1-independent growth, it has little-to-no impact on the behaviour of intestinal epithelium in organoid culture. This work highlights that the outcome of endogenous chromosome rearrangements can differ significantly from cDNA overexpression (compare Fig. 3, Supplementary Figs 1 and 5c), and that Rspo2 and Rspo3 rearranged cells are molecularly distinct from those carrying Apc alterations.

**Rspo rearrangements initiate tumour growth in vivo.** To directly test whether Rspo rearrangements are sufficient to drive a phenotypic response in vivo, we treated R26-rtTA/c3GCI9-E-Rspo2 or R26-rtTA/c3GCI9-P-Rspo3 bi-transgenic mice with doxycycline for 10 days and examined mice at various times. Two weeks following Cas9 induction, the intestine of c3GIC9-E-Rspo2 animals appeared largely normal with a few areas of crypt-villus disruption and minimal hyperproliferation. However, by 6 weeks there were numerous hyperproliferative and dysplastic lesions, that appeared histologically similar to those produced following Cas9-mediated Apc disruption[11] (Fig. 4a; Supplementary Fig. 11b). Interestingly, c3GIC9-E-Rspo2 animals examined 20–25 weeks post Cas9 induction showed near-identical sized lesions (Supplementary Figs 11b and 12), indicating that while the E-Rspo2 rearrangement can initiate small hyperproliferative adenomas, its effect alone is not sufficient to promote continued tumour growth.

Mirroring the increased potency in organoids, P-Rspo3 mice showed clear intestinal lesions at 2 weeks that progressed to widespread hyperplastic and dysplastic adenomas by 5 weeks (Fig. 4a). In fact, at this timepoint, much of the duodenum and jejunum appeared disordered, with minimal remaining normal epithelium (Supplementary Fig. 11a). More distal regions of the intestine showed a less marked transformation, although there were obvious areas of hyperproliferation and dysplasia. In contrast to what we observed in organoid culture, both E-Rspo2 and P-Rspo3 rearrangements drove hyperproliferation (Ki67) and showed a profound lack of epithelial differentiation, as measured by Krt20 and alkaline phosphatase staining (Fig. 4a; Supplementary Fig. 11b). Similar to Apc-mutant tumours, we observed ectopic production of Paneth cells (Lysozyme[+]) throughout the lesions (Fig. 4a; Supplementary Fig. 11b). DNA FISH analyses on intestinal sections confirmed that the E-Rspo2 and P-Rspo3 genomic rearrangements were present throughout all hyperproliferative lesions and adenomas examined (n = 15 tumours/genotype) (Fig. 4b, Supplementary Fig. 11c). Moreover, analysis of bulk intestinal tissue from P-Rspo3 mice, 8 days and 5 weeks following Cas9 induction, showed a marked enrichment of cells containing the genomic P-Rspo3 rearrangement (Fig. 4c). By 5 weeks, almost 30% of the intestine contained P-Rspo3-positive cells, and this was reflected by a significant increase in Rspo3 transcript (Fig. 4c, n = 5–11, P < 0.0001). In addition, sub-cutaneous transplantation of P-Rspo3-positive organoids

revealed that this rearrangement is capable of supporting tumorigenic growth outside the intestinal environment, similar to Apc loss-of-function (Supplementary Fig. 13). While we cannot rule out the possibility that Rspo3 secretion from hyperplastic epithelial cells, or Rspo3 rearrangements in surrounding stromal cells, can alter the behaviour of normal epithelium, our data suggest that cell-intrinsic proliferative expansion of P-Rspo3-positive cells is the primary mechanism driving tumour initiation and growth.

Our in vivo data suggest that Rspo rearrangements drive acute hyperproliferation and tumour initiation in the intestine similar to mutational loss of Apc. Yet, our ex vivo transcriptome analysis highlighted marked gene expression differences between Apc-mutant and P-Rspo3 organoids. To determine whether differences identified in organoids were a true reflection of P-Rspo3 and Apc-mutant adenomas in vivo, we examined the expression of several differentially regulated genes by immunohistochemistry. Sox17 is a marker of definitive endoderm during embryogenesis, and was the most upregulated gene in Apc organoids (Fig. 5a; Supplementary Data 1). Consistent with this, Apc-mutant adenomas induced by CRISPR or Cre showed clear and abundant nuclear Sox17 staining in intestinal lesions (Fig. 5b; Supplementary Fig. 14). In contrast, Sox17 was completely absent from normal intestinal crypts and could not be detected in P-Rspo3 tumours (Fig. 5b), accurately reflecting the gene expression observed in organoids. Sox9 is a transcription factor mutated and overexpressed in a subset of human CRC, and upregulated in Apc-mutant organoids (Fig. 5a). In vivo, Sox9 was markedly induced in Apc-mutant adenomas, but showed limited and mosaic expression in P-Rspo3 lesions, mirroring normal intestinal crypts (Fig. 5b; Supplementary Fig. 14). Similarly, Axin2, a classic Wnt target gene, was markedly upregulated in Apc-mutant, but not P-Rspo3, organoids or adenomas (Fig. 5; Supplementary Fig. 14). Thus, the molecular characterization of Apc[mut] and P-Rspo3 intestinal lesions mirrors the gene expression profiles identified in organoids, and indicates that while Apc mutations and Rspo rearrangements drive the development of morphologically similar intestinal tumours, each genetic event induces a distinct molecular profile.

**Rspo rearranged tumours are sensitive to Porcn inhibition.** Consistent with their transcriptional and morphological similarity to wildtype organoids, P-Rspo3 cultures were absolutely dependent on Wnt ligand-mediated signalling, as inhibition of Wnt secretion by Porcupine (PORCN) inhibitors (C59 and LGK974) blocked proliferation and crypt budding (Fig. 6a,b). These data are consistent with recent work showing that PORCN inhibition can stall tumour growth in sub-cutaneous CRC xenografts[26]. However, we reasoned that our CRISPR-based model offered a refined context to examine therapeutic response, eliminating issues of species-selective drug potency, and providing an autochthonous setting to define efficacy and toxicity. Moreover, as wildtype and P-Rspo3 organoids were equally sensitive to Wnt blockade, we sought to determine whether there is a realistic therapeutic window for treatment of Rspo-fusion tumours growing within the normal intestinal mucosa. To do this, we treated R26-rtTA/E-Rspo2 and R26-rtTA/P-Rspo3 mice with dox for 10 days, allowed intestinal lesions to develop over 6 weeks, and randomized the animals for treatment with either vehicle (DMSO) or LGK974 (5 mg kg$^{-1}$) for 7 days (Fig. 6c). Only 1 week of LGK974 treatment completely abolished E-Rspo2 lesions in the small intestine, but had no apparent effect on the morphology, differentiation or proliferation of stem and progenitor cells in the normal intestinal crypts of tumour bearing mice, or wildtype control

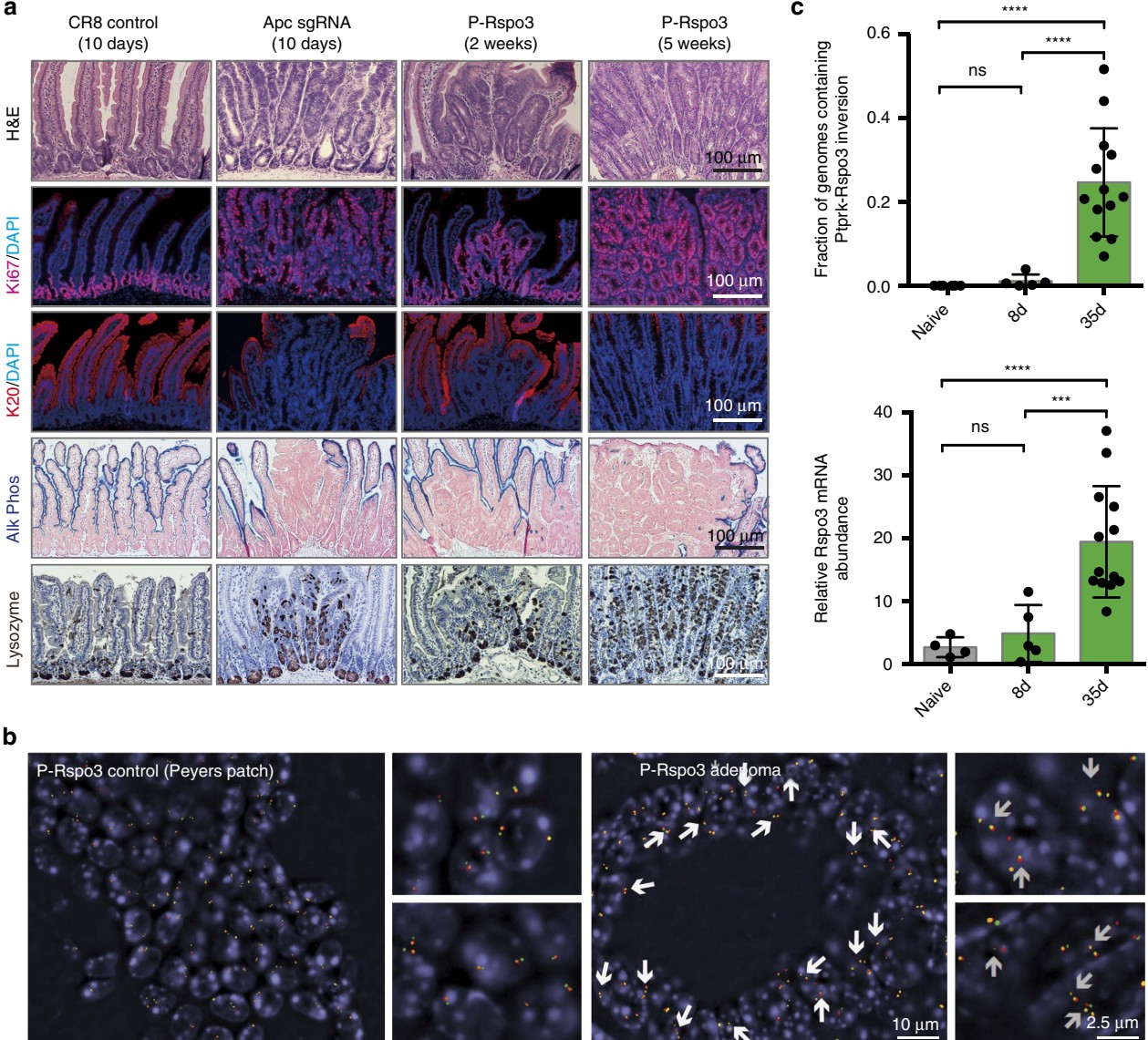

**Figure 4 | Rspo rearrangements initiate tumour growth *in vivo*.** (**a**) Immunohistochemical (H&E, alkaline phosphatase and lysozyme), and immunofluorescent (Ki67 and keratin 20) stains of intestinal sections from control sgRNA (*CR8*), *Apc* sgRNA and *P-Rspo3* animals, treated with dox (200 mg kg$^{-1}$) for 10 days, and collected at indicated times. Scale bars, 100 μm. (**b**) DNA FISH staining on intestinal sections from *R26-rtTA/c3GIC9-P-Rspo3* animals. Immune cells within Peyers patches are shown as a normal control. P-Rspo3 adenomas are enriched for the *P-Rspo3* rearrangement, highlighted with white arrows. (**c**) Graphs represent fraction of genomes containing *P-Rspo3* inversions (upper) and mRNA expression level of Rspo3 (lower) in mouse small intestine following dox treatments for indicated times ($n \geq 4$, bars represent mean values +/− s.d., ***$P < 0.001$, ****$P < 0.0001$, two-sided *t*-test with Welch correction).

animals (Fig. 6c; Supplementary Fig. 15). Remarkably, while *R26-rtTA/P-Rspo3* mice had markedly increased tumour burden, they showed a near-complete regression following only 7 days of LGK treatment (Fig. 6d). LGK-treated mice showed significantly reduced proliferation outside the crypt (Fig. 6e), and while histologically abnormal due to rapid regression of the intestinal tumours, villi in *P-Rspo3* mice showed a normal pattern of differentiation (Supplementary Fig. 15c). Thus, pharmacologic inhibition of WNT ligand production drives a rapid differentiation response in *Rspo* rearranged tumours, but not normal crypts.

Almost all *RSPO2* and *RSPO3* rearranged human CRCs carry activating mutations in *KRAS* or *BRAF*, and many show loss-of-function mutations in *TP53* and *SMAD4* (refs 7,27). To extend our treatment studies to more complex genotypes, we

used an established Cre-dependent *LSL-Braf*$^{V618E}$ mouse strain[28,29], and *ex vivo* CRISPR-based editing to generate *Braf*$^{V618E}$/*P-Rspo3* (*BR3*) organoids (Fig. 6f; Supplementary Fig. 16). Further, through sequential CRISPR-mediated mutagenesis, we generated *Braf*$^{V618E}$/*P-Rspo3/Smad4/Trp53* (*BRPS*) quadruple mutant cultures (Fig. 6f; Supplementary Fig. 16) that grew as large proliferative spheres, reminiscent of *Apc*-mutant cultures. The presence of an oncogenic Braf mutation did not alter the response to LGK974, as *BR3* organoids showed complete cell cycle arrest within 4 days of treatment, similar to *WT* and *P-Rspo3* cells (Fig. 6g). While *BRPS* cells did not display an immediate morphological response to LGK974, they showed a profound decrease in proliferation and coincident upregulation of differentiation (Krt20) after 4 days of treatment (Fig. 6g).

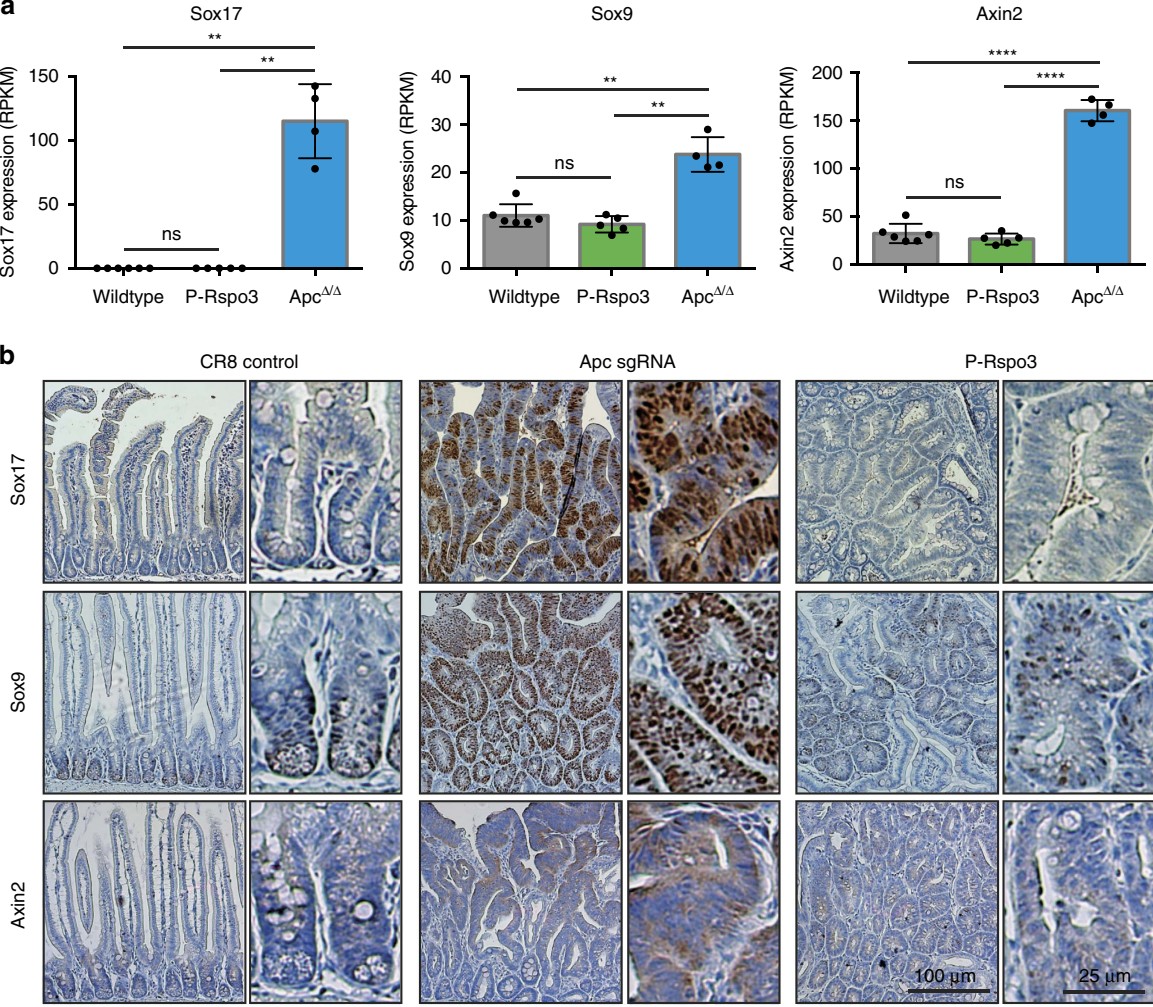

**Figure 5 | P-Rspo3 tumours are molecularly distinct from Apc-mutant tumours. (a)** Graphs represent expression (RPKM value) of Sox17, Sox9 and Axin2 on WT, P-Rspo3 and Apc-deleted organoids ($n \geq 4$, bars represent mean values $+/-$ s.d., $**P < 0.01$, $***P < 0.001$, $****P < 0.0001$, two-sided $t$-test with Welch correction). **(b)** Immunohistochemical staining of Sox17, Sox9 and Axin2 on small intestinal sections from dox-treated CR8, Apc sgRNA and P-Rspo3 animals. Scale bars are labelled in the figure.

Altogether, our studies suggest that *E-Rspo2-* and *P-Rspo3-* driven tumours are exquisitely sensitive to Porcn inhibitors *in vivo*, and that even in the presence of additional oncogenic insults, these cells remain dependent on WNT signalling for proliferative expansion.

## Discussion

Here, we describe the first application of transgenic and inducible CRISPR/Cas9 technology to investigate cancer-associated chromosome rearrangements *in vivo*, and show that endogenous *PTPRK–RSPO3* and *EIF3E–RSPO2* rearrangements are sufficient to initiate hyperproliferation and tumour development in the intestine. *Rspo*-driven hyperplastic and dysplastic polyps look indistinguishable from *Apc*-mutant adenomas, yet they are molecularly distinct. While both genomic events induce upregulation of Rspo—a potent Wnt pathway agonist—they do not induce broad transcriptional changes characteristic of *Apc* disruption. Recently, Bakker and colleagues[23] described a Cre-dependent *Rspo3* transgene model, whereby the (non-fused) Rspo3 cDNA is expressed in intestinal stem cells following induction of Cre in Lgr5-positive cells. In this system, they reported a moderate upregulation of classic Wnt pathway

targets, such as *Lgr5* and *Axin2* in the intestine, spheroid changes in organoids, and a tumour response driven predominantly by paracrine Rspo3 signals from a minor sub-population of cells. While we show that *P-Rspo3* rearrangements can maintain Wnt target gene expression in the absence of exogenous RSPO1 ligand (Supplementary Fig. 6), we did not detect changes in the vast majority of Wnt targets. In fact, we saw almost no change in the transcriptional profile of *P-Rspo3* organoids, compared to wildtype cells cultured in RSPO1. Likewise, we observed no effects in organoid culture. Although this lack of response *in vitro* is at odds with the profound tumorigenic response *in vivo*, we speculate that the production of excess Rspo3 simply allows *P-Rspo3* fusion cells to maintain their stem and progenitor cell phenotype independent of the crypt niche. This idea is supported by the observation that hyperplastic and dysplastic adenomas *in vivo* look molecularly similar to normal intestinal crypts (Fig. 5).

It is not clear what underlies the difference between our model and that reported by Hilkens *et al.*, but it may be in the nature of *Rspo3* expression. Hilkens *et al.* used a synthetic CAGs promoter to drive ectopic expression of *Rspo3*, while our approach relied on expression from the endogenous *Ptprk* promoter, and induction of a *PTPRK–RSPO3* fusion protein. Indeed, we observed a clear

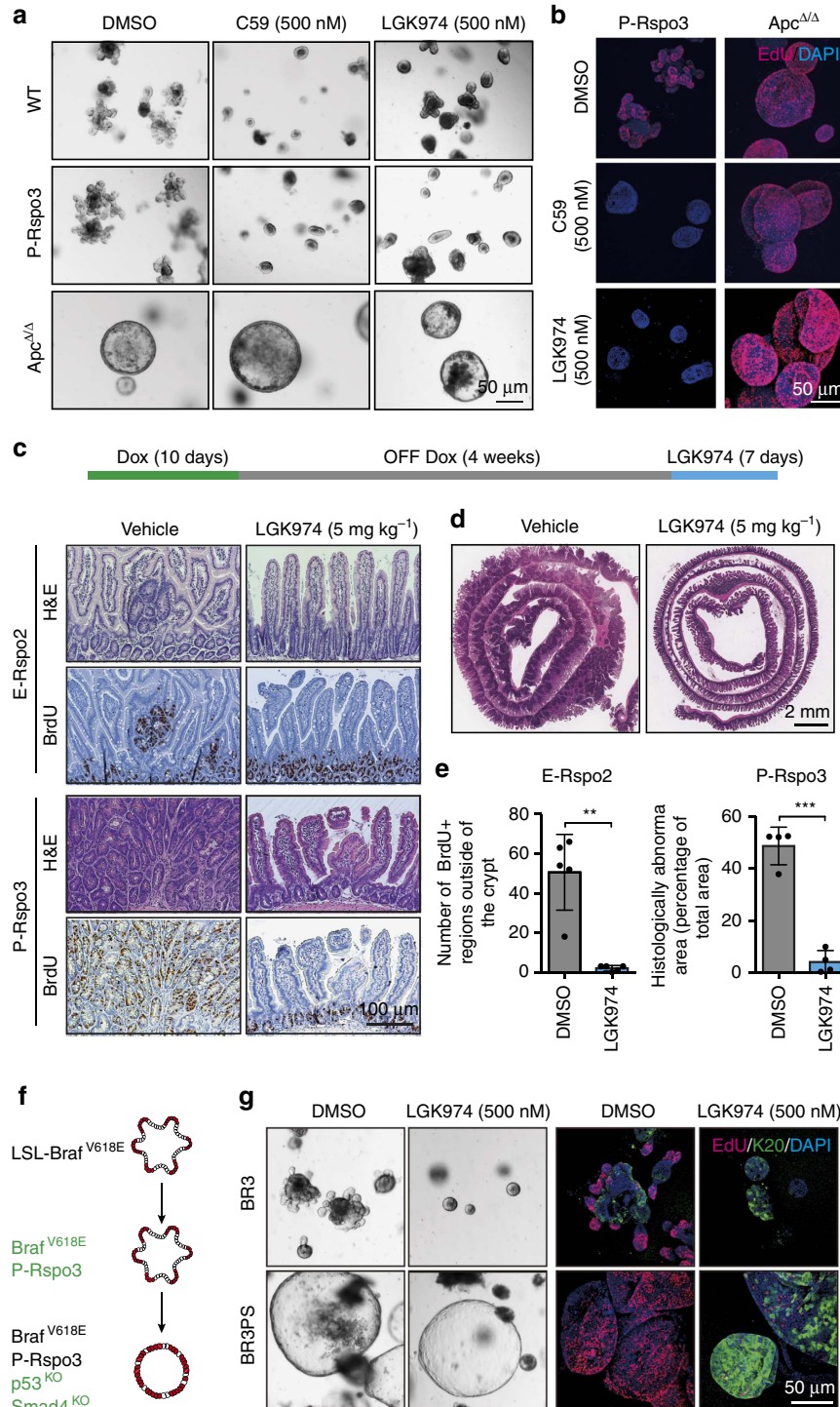

**Figure 6 | Rspo rearranged tumours are sensitive to Porcn inhibition.** (**a**) Bright-field images of WT, P-Rspo3, and Apc-deleted organoids treated with DMSO, C59 (500 nM) or LGK974 (500 nM) for 4 days. Scale bars, 50 µm. (**b**) Immunofluorescent images of P-Rspo3 and Apc-deleted organoids treated with DMSO, C59 (500 nM), and LGK974 (500 nM) for 4 days. EdU (red) was stained for proliferation. Scale bars, 50 µm. (**c**) Schematic of the *in vivo* LGK974 treatment experiment. (**c**) Immunohistochemical images of intestinal sections from E-Rspo2 and *R26-rtTA/P-Rspo3* mice treated with either DMSO or LGK974. BrdU was stained for proliferation. Scale bars, 100 µm. (**d**) Representative immunohistochemical images of proximal small intestine from P-Rspo3 mice treated with either DMSO or LGK974. (**e**) Quantification of hyperproliferative and histologically abnormal regions in the LGK974 treatment experiment (*n* ≥ 4, bars represent mean values + / − s.d., **P < 0.01, ***P < 0.001, two-sided *t*-test with Welch correction). (**f**) Schematic representation of sequential gene editing in intestinal organoids to create BR3 and BRPS cultures. Red indicates proliferative cells. (**g**) Bright-field and immunofluorescent images of DMSO or LGK974 treated, BR3 and BRPS organoids, as indicated. In both genotypes, 4 days of LGK treatment induce cell cycle arrest and drives differentiation. Scale bars, 50 µm.

dose-dependent phenomenon in the case of Rspo2; whereas enforced overexpression of the *Rspo2* cDNA enabled RSPO1-independent proliferation in organoids, the endogenous *EIF3E–* *RSPO2* rearrangement could not. In addition, while we do not expect it to contribute significantly to the initial Rspo3-mediated response, it is possible that the reciprocal loss of one copy of

*Ptprk* may influence tumour progression, as it has previously been reported as a putative tumour suppressor in animal models of intestinal cancer[30]. Inducible CRISPR systems like those described here, provide a powerful tool to engineer the precise genetic events observed in human cancer and produce high-fidelity disease models for pre-clinical evaluation.

Owing to the Wnt-dependent mechanism of RSPO function[8], *RSPO*-fusion CRCs represent a subclass of cancers that may be treated with Wnt-targeted agents. Indeed, initial studies using sub-cutaneous xenografts of RSPO-fusion human CRCs, highlighted the potential of PORCN inhibitors[26] or direct RSPO3 neutralizing antibodies[31] for treatment of these tumours. Similarly, Clevers and colleagues[32] showed evidence that human RNF43 mutant organoids are sensitive to PORCNi. Our data, using an autochthonous *in vivo* model system, suggest an exquisite sensitivity of Rspo-fusion tumours to PORCNi, relative to normal intestinal crypts. This hypersensitivity of proliferative tumour cells and relative protection of normal intestinal crypts could not have been predicted from *ex vivo* organoid experiments, and highlights the power of the *in vivo* models for studying therapeutic response. Exactly what dictates the differential response to LGK974 in not known, though it is plausible that the resilience of normal crypt cells comes from their position within the stem-cell niche. Close interaction with supporting cells such as pericryptal myofibroblasts, that may provide supplemental growth factor support required following PORCN inhibitor treatment (for review see ref. 33). Hence, while our data suggest PORCN inhibitors are potent anti-tumour drugs, and we expect there is likely a genuine therapeutic window for treatment of *RSPO*-fusion tumours, further work will be required to determine whether tumour cells can evade elimination, as do normal crypt stem and progenitor cells.

The initial finding that *RSPO* fusions are mutually exclusive with *APC* mutations in human CRC implied that these events have redundant effects in tumour development. Our study shows that indeed, endogenous *RSPO2* and *RSPO3* rearrangements can initiate tumour growth in the intestine, but that they drive disease on a molecularly distinct course compared to *Apc* mutations. Most importantly, both *Rspo2* and *Rspo3* fusions represent druggable cancer drivers, as the tumours are sensitive to Wnt-targeted therapy with a meaningful therapeutic window. Finally, our data highlight the utility of transgenic CRISPR models to simply and quickly engineer large-scale chromosome alterations, and we expect these systems will be invaluable for interrogating the impact of newly identified, cancer-associated genetic variants, and developing tailored pre-clinical models.

## Methods

**Cloning.** Sequences encoding guide RNAs (Supplementary Data 5) were cloned into PX458 for initial validation[34]. U6 promoter and guide RNA cassettes were PCR-amplified (sequences in Supplementary Data 5) and cloned (NsiI/SbfI) into an NsiI site upstream of the TRE3G promoter of c3GIC9 vector[11].

**ES cell targeting.** KH2 ES cells were maintained on irradiated feeders in M15 media containing LIF as previously outlined[3]. Two days following transfections cells were treated with media containing 150 µg ml$^{-1}$ hygromycin and individual surviving clones were picked after 9–10 days of selection. Two days after clones were picked hygromycin was removed from the media and cells were cultures in standard M15 thereafter. To confirm single copy integration at the col1a1 locus, we first validated expected integration by multiplex col1a1 PCR[3], and second, confirmed the presence of a single GFP cassette using the Taqman copy number assay, according to the manufacturer's instructions (Invitrogen).

**Animal studies.** ES cell-derived mice were produced by blastocyst injection and animals were maintained on a mixed C57B6/129 background. Progeny of both sexes were used for experiments and were genotyped for specific alleles (*Lgr5-CreER, Apcflox, R26-rtTA* and *col1a1*) using primers and protocols previously described[6,11]. Production of mice and all treatments described were approved by the Institutional Animal Care and Use Committee (IACUC) at Weill Cornell Medicine (NY), under protocol number 2014-0038. Doxycycline was administered via food pellets (200 mg kg$^{-1}$) (Harlan Teklad) at 5–6 weeks of age. Animals were removed from doxycycline diet after 10 days and collected at various time points, detailed in the Results section. For LGK974 treatment studies, LGK974 (5 mg kg$^{-1}$, Selleckchem #S7143) was mixed with 0.5% methylcellulose and 0.1% Tween 80 and then administrated by daily oral gavage. Animals were weighed everyday during treatment and the mice were killed after 7 days. Animal studies were not blinded during treatment, however, quantitation of tumour burden involved measurements by two parties, one blinded to the treatment groups.

**Immunohistochemistry and immunofluorescence.** Tissue, fixed in freshly prepared 4% paraformaldehyde for 24 h, was embedded in paraffin and sectioned by IDEXX RADIL (Columbia, MO, USA). Sections were rehydrated and unmasked (antigen retrieval) by either: (i) Heat treatment for 5 min in a pressure cooker in 10 mM Tris/1 mM EDTA buffer (pH 9) containing 0.05% Tween 20 or (ii) Proteinase K treatment (200 µg ml$^{-1}$) for 10 min at 37 °C (lysozyme staining). For immunohistochemistry, sections were treated with 3% H$_2$O$_2$ for 10 min and blocked in TBS/0.1% Triton X-100 containing 1% BSA. For immunofluorescence, sections were not treated with peroxidase. Primary antibodies, incubated at 4 °C overnight in blocking buffer, were: rabbit anti-ki67 (1:100, Sp6 clone, Abcam #ab16667), rabbit anti-KRT20 (1:200, Cell Signaling Technologies, #13063), rabbit anti-Lysozyme (1:400, Dako, #EC 3.2.1.17), rat anti-BrdU (1:200, Abcam #ab6326), rabbit anti-Sox17 (1:200, R&D, #AF1924), rabbit anti-Sox9 (1:1,000, Millipore #AB5535), rabbit anti-Axin2 (1:800, Abcam #ab32197). For immunohistochemistry, sections were incubated with anti-rabbit or anti-rat ImmPRESS HRP-conjugated secondary antibodies (Vector Laboratories, #MP7401) and chromagen development performed using ImmPact DAB (Vector Laboratories, #SK4105). Stained slides were counterstained with Harris' hematoxylin. For immunofluorescent stains, secondary antibodies were applied in TBS for 1 h at room temp in the dark, washed twice with TBS, counterstained for 5 min with DAPI and mounted in ProLong Gold (Life Technologies, #P36930). Secondary antibodies used were: anti-rabbit 568 (1:500, Molecular Probes, #a11036). Images of fluorescent and IHC stained sections were acquired on a Zeiss Axioscope Imager (chromogenic stains), Nikon Eclipse T1 microscope (IF stains). Raw.tif files were processed using FIJI (Image J) and/or Photoshop (Adobe Systems, San Jose, CA, USA) to create stacks, adjust levels and/or apply false colouring.

**Isolation and culture of intestinal organoids.** Isolation, maintenance and staining of mouse intestinal organoids has been described previously[35,36]. Briefly, for isolation, 15 cm of the proximal small intestine was removed and flushed with cold PBS. The intestine was then cut into 5 mm pieces, vigorously resuspended in 5 mM EDTA-PBS using a 10 ml pipette, and placed at 4 °C on a benchtop roller for 10 min. This was then repeated for a second time for 30 min. After repeated mechanical disruption by pipette, released crypts were mixed with 10 ml DMEM Basal Media (Advanced DMEM F/12 containing Pen/Strep, Glutamine, 1 mM N-Acetylcysteine (Sigma Aldrich A9165-SG)) containing 10 U ml$^{-1}$ DNAse I (Roche, 04716728001), and filtered sequentially through 100 and 70 µm filters. 1 ml FBS (final 5%) was added to the filtrate and spun at 125 g for 4 min. The purified crypts were resuspended in basal media and mixed 1:10 with Growth Factor Reduced Matrigel (BD, 354230). Overall, 40 µl of the resuspension was plated per well in a 48-well plate and placed in a 37 °C incubator to polymerize for 10 min. Also, 250 µl of small intestinal organoid growth media (Basal Media containing 50 ng ml$^{-1}$ EGF (Invitrogen PMG8043), 100 ng ml$^{-1}$ Noggin (Peprotech 250-38) and 500 ng ml$^{-1}$ R-spondin (R&D Systems, 3474-RS-050, or from conditioned media) was then laid on top of the Matrigel. Where indicated, dox was added to experiments at 500 ng ml$^{-1}$.

For sub-culture and maintenance, media was changed on organoids every 2 days and they were passaged 1:4 every 5–7 days. To passage, the growth media was removed and the Matrigel was resuspended in cold PBS and transferred to a 15 ml falcon tube. The organoids were mechanically disassociated using a p1000 or a p200 pipette, and pipetting 50–100 times. Overall, 7 ml of cold PBS was added to the tube and pipetted 20 times to fully wash the cells. The cells were then centrifuged at 125 g for 5 min and the supernatant was aspirated. They were then resuspended in GFR Matrigel and replated as above. For freezing, after spinning the cells were resuspended in Basal Media containing 10% FBS and 10% DMSO and stored in liquid nitrogen indefinitely.

**Organoid imaging.** For live imaging, organoids were cultured in 24-well plates and imaged in a Nikon Biostation (37C, 5% CO$_2$) every hour, over 3 days. Movies were constructed by compiling individual TIF files in FIJI (Image J). For fixed staining, organoids were grown in 40 µl of Matrigel plated into an 8-well chamber slide (Lab-Tek II, 154534). Where indicated, 10 µM EdU was added to the growth media for 6 h before fixing. The growth media was removed and the cells were fixed in 4% PFA-PME (50 mM PIPES, 2.5 mM MgCl$_2$, 5 mM EDTA) for 20 min. They were then permeabilized in.5% Triton for 20 min and blocked in IF Buffer (PBS, 0.2% Triton, 0.05% Tween, 1% BSA) for 1 h or immediately processed for EdU staining performed according to directions provided with the Click-iT Edu Alexa Fluor 647 Imaging Kit (Invitrogen C10340). For alkaline phosphatase staining,

fixed cells were washed twice with TBS and then incubated with the BCIP/NBT Substrate Kit (Vector Laboratories, SK-5400) for 15 min in the dark. The chambers were then washed twice with TBS and then imaged using bright-field microscopy. For immunofluorescent staining, cells were incubated in primary antibodies overnight in IF buffer: rabbit anti-KRT20 (1:200, Cell Signaling Technologies, #13063), rabbit anti-Lysozyme (1:200, Dako, #EC 3.2.1.17). They were then washed three times with TBS.1% Tween. Secondary antibodies (1:1,000, same reagents as above) were incubated with or without Alexa-647 Phalloidin (Molecular Probes, #A22287) for 1 h. The solution was removed and DAPI in PBS was added for 5 min and washed twice with TBS 0.1% Tween. The chambers were then removed and cover slips were mounted using Prolong Gold antifade medium (Invitrogen P36930). Images were acquired using Zeiss LSM 880 laser scanning confocal microscope, and Zeiss image acquiring and processing software. Images were processed using FIJI (Image J) and Photoshop CS5 software (Adobe Systems, San Jose, CA, USA).

**Organoid transfection.** Murine small intestinal organoids were cultured in transfection medium containing CHIR99021 (5 μM) and Y-27632 (10 μM) for 2 days prior to transfection. Single cells suspensions were produced by dissociating organoids with TrypLE express (Invitrogen #12604) for 5 min at 37 °C. After trypsinization, cell clusters in 300 μl transfection medium were combined with 100 μl DMEM/F12-lipofectamine 2000 (Invitrogen #11668)-DNA mixture (97 μl–2 μl–1 μg), and transferred into a 48-well culture plate. The plate was centrifuged at 600 g at 32 °C for 60 min, followed by another 6 h incubation at 37 °C. The cell clusters were spun down and plated in Matrigel. For selecting organoids with *PTPRK–RSPO3* rearrangements, exogenous R-spondin1 was withdrawn 1 week after transfection. Organoids were cultured in medium containing Nutlin3 (5 μM) and TGFβ1 (5 ng ml$^{-1}$) for 1 week to select for *p53* loss and *Smad4* loss.

**Protein analysis.** Small intestine organoids were grown in 100 μl of Matrigel in four wells each of a 12-well dish for 4 days post-passage. Organoids were then recovered from the Matrigel using several rinses with cold PBS. Organoid pellets were lysed with RIPA buffer. Antibodies used for western blot were: anti-R-spondin3 (Proteintech #17193-1-AP) and anti-actin-HRP (Abcam #ab49900).

**Flow cytometry.** TdTomato protein abundance was measured by calculating mean fluorescence intensity, following analysis on a BD Accuri C6 flow cytometer. Experiments described represent three independent viral transductions, each at different multiplicity of infection (MOI), to account for any effect of gene dosage.

**RNA isolation and qPCR.** RNA was extracted using TRIzol according to the manufacturer's instructions and contaminating DNA was removed by DNase treatment for 10 min and column purification (Qiagen RNeasy). cDNA was prepared from 1 μg total RNA using qScript reverse transcription kit (Quantabio, #95047). Quantitative PCR detection was performed using SYBR green reagents (Applied Biosystems) and specific primers listed in Supplementary Data 5.

**Genomic DNA Taqman assay.** For quantification of *PTPRK–RSPO3* inversions in complex tissue and organoids: Taqman PCR was conducted on QuantStudio 6 Real-Time PCR system (Applied biosystems), using Taqman master mix reagent (Applied biosystems) and specific primers and probe (Supplementary Data 5).

**RNAseq analysis.** The quality of the raw FASTQ files were checked with FastQC. Raw FASTQ reads were mapped to mouse reference GRCm38 using STAR two-pass alignment (v2.4.1d; default parameters), estimated gene expression using cufflinks (v2.2.1), and quantified per-gene counts using HTseq (v0.6.0)[37–39]. An FPKM threshold of 0.1 was set to define expressed genes and used the DESeq2 variance stabilized transform function on the read counts for downstream unsupervised analyses[40]. To confirm the presence of the PTPRK–RSPO3 gene fusion we mapped reads to GRCm38 using STAR chimeric alignment, extracted reads that mapped +1/−1 base pair around the expected junction location, realigned to the fused and wildtype sequences and visualized with IGV[41]. We additionally examined the read counts mapping the RSPO3 and PTPRK exons to confirm the fusion. To confirm that APC group models transcriptional profiles in colorectal cancer we extracted all gene sets from the C2 MSigDB database that contain either the keyword 'colorectal' or 'colon'[42]. We then performed GSEA pre-ranked analysis (v2.2.1) on the log$_2$ fold change calculated after testing differential expression between the APC-mutant and APC-WT groups. To provide additional validation, we developed a gene signature of colorectal cancer (COAD) using TCGA data downloaded from Firehouse. The downloaded data included 459 COAD samples and 41 normal control samples. We tested differential expression between normal and cancer samples to develop a 1158 gene signature (adjusted *P*-value of 0.01 and log$_2$ fold change of 3). We again used GSEA pre-ranked on the log$_2$ fold changes between APC-mutant and APC-WT to confirm enrichment of our COAD gene signature. We used R (v3.2.2) and the *gplots* package to create all visualizations and to perform hierarchical clustering and principal component

analysis. We used the vertebrate homology list provided by Mouse Genome Informatics (ftp://ftp.informatics.jax.org/pub/reports/HOM_MouseHuman Sequence.rpt) to convert mouse gene symbols to human gene symbols. We used DESeq2 to perform all differential expression analyses (v1.12.3)[40], and utilized Nextflow (http://dx.doi.org/10.6084/m9.figshare.1254958) to implement our computational pipelines.

**Mutation detection by T7 assay.** Cas9-induced mutations were detected using the T7 endonuclease I (NEB). Briefly, an ∼500 bp region surrounding the expected mutation site was PCR-amplified using Herculase II (600675, Agilent Technologies) (primer sequences in Supplementary Data 5). PCR products were column purified (Qiagen) and subjected to a series of melt-anneal temperature cycles with annealing temperatures gradually lowered in each successive cycle. T7 endonuclease I was then added to selectively digest heteroduplex DNA. Digest products were visualized on a 2.5% agarose gel. Uncropped images of gels are shown in Supplementary Fig. 17.

**Data availability.** Raw RNAseq data have been deposited in the Genbank GEO database (http://www.ncbi.nlm.nih.gov/geo/) under accession codes GSE98257. All other remaining data are available within the article and supplementary files, or available from the authors upon request.

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

## Acknowledgements

We thank Ashlesha Muley and Sukanya Goswami for technical assistance with animal colonies, Arun Unni and Miguel Foronda for critical reading of the manuscript, and other members of the Dow laboratory for advice and discussions. We thank Cameron Nowell, Research Facilities Manager, Monash Institute of Pharmaceutical Science, Melbourne, Australia, for the development of a computational script for exporting whole-slide scans. Whole-slide scans were performed in collaboration with Institute for Precision Medicine and Meyer Cancer Center. Microscopy was performed at the Weill Cornell Medicine Optical Microscopy Core Facility, and RNAseq was conducted at the Weill Cornell Medicine Genomics Core Facility. DNA FISH analysis was performed by the Memorial Sloan Kettering Cancer Center (MSKCC) Molecular Cytogenetics Core, with support from an NIH Cancer Center support grant (P30 CA008748). This work was supported by a pilot grant from the Center for Advanced Digestive Care (CADC) at Weill Cornell Medicine, and a project grant from the NIH/NCI (CA195787-01) and a project grant from the Starr Cancer Consortium (I10-0095). E.M.S. was supported by a Medical Scientist Training Program grant from the National Institute of General Medical Sciences of the National Institutes of Health under Award Number T32GM07739 to the Weill Cornell/Rockefeller/Sloan-Kettering Tri-Institutional MD-PhD Program. L.E.D. was supported by a K22 Career Development Award from the NCI/NIH (CA 181280-01).

## Author contributions

T.H. performed experiments, analysed data and wrote the paper. E.M.S. and M.P.Z. performed experiments and analysed data. C.M. and O.E. performed data analysis. L.E.D. conceived the project, performed and supervised experiments, analysed data and wrote the paper.

## Additional information

**Competing interests:** The authors declare no competing financial interests.

