## [Peer Review File · Nature Communications]

REVIEWERS' COMMENTS:

Reviewer #1 (Remarks to the Author):

The authors of the manuscript have described the effect of RSPO chromosome rearrangement in the context of Wnt-dependent tumour initiation using in vivo mouse model. In the revised manuscript, the authors tried to address the questions raised by the reviewers with solid logic and evidence. Moreover, this finding is important and the manuscript broadens our understanding about Wnt-dependent tumour initiation. Therefore, the revised version of the manuscript improved the quality and deserves publication in Nature Communication.

Reviewer #2 (Remarks to the Author):

The authors have addressed all my comments appropriately and nicely improved the quality of the manuscript. I believe it is now appropriate for publication in Nature Communication.

Reviewer #3 (Remarks to the Author):

In the revised version of their manuscript, Han et al. have attempted to answer most of the concerns raised by me and the other reviewers.

I believe that their new data have strengthened the previous data and showed more evidences supporting their conclusions. However, in my opinion, the paper remains rather descriptive. The main claim of the work – the Ptpk-Rspo3 fusions provide a cell-intrinsic growth advantage – it is now more convincingly shown, but still this interesting observation is not supported by mechanistic data showing how this happens.

At this point I believe it is more an editorial decision whether the conclusions of the paper can guarantee publication in Nature Communication.